# Identifying Sustainable Grassland Management Approaches in Response to the Invasive Legume *Lespedeza cuneata*: A Functional Group Approach

**Erin M. Garrett**[1] **and David J. Gibson**[2,*] 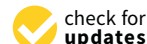

[1]   University of Illinois Extension, University of Illinois, Champaign, IL 61820, USA; emedvecz@illinois.edu
[2]   School of Biological Sciences, Southern Illinois University, Carbondale, IL 62901, USA
*   Correspondence: djgibson@siu.edu; Tel.: +1-618-453-3231

**Abstract:** We propose combining the filter framework model of community assembly with the passenger-driver model of non-native species behavior to help clarify the impacts of invasive species in the communities they invade and to guide sustainable management protocols. Observational field surveys and a greenhouse experiment explored the role of the invasive legume *Lespedeza cuneata* in the communities it invades and how natives in three functional groups—grasses, forbs, and legumes—respond to its presence. Within-site analyses from the field survey revealed differences in invaded and uninvaded areas in half of the sites, suggesting that site-specific characteristics influences whether *L. cuneata*'s presence corresponds to local differences in species composition. The greenhouse experiment found higher levels of saprophytic and arbuscular mycorrhizal fungi in soil conditioned by *L. cuneata* than in unconditioned soil. However, competition between *L. cuneata* or the native congener *L. capitata* and nine native species illustrated stronger aboveground competitive effects than belowground soil effects due to soil conditioning, with impacts differing among functional groups. The response of *L. cuneata* was reduced in the presence of grasses and other legumes but not forbs. Assessing the impact of *L. cuneata* with the combined community assembly model revealed this invasive plant acts as a driver because it alters abiotic and biotic filters to impact species composition. Managing for high grass abundance and planting native legumes will help sustain grasslands from *L. cuneata* invasion.

**Keywords:** competition; filter model; grassland; *Lespedeza cuneata*; passenger-driver model; plant-soil feedback; sustainable management

## 1. Introduction

Studies of non-native invasive species dynamics need to include invasive impacts on community assembly, especially when the control of the invasive species through management cannot keep pace with its spread. A relevant model of community assembly for studying plant invasions is the filter framework model, in which biotic and abiotic filters determine the species composition of an ecosystem by only allowing species able to "pass through" the filters to establish and thrive [1–5]. Biotic filters are constraints imposed by the living components of an ecosystem, such as competition, predation, and mutualisms, while abiotic filters are limiting environmental and climatic factors, such as soil composition, water availability, and temperature. By contrast, the passenger-driver model of community assembly differentiates between two types of non-native species: drivers, which are able to establish and dominate in an ecosystem, and passengers, which establish themselves but do not adversely affect other species [6,7]. Approximately 10% of non-native species act as drivers, lowering community diversity and creating novel communities of plants through competition, soil conditioning,

withstanding disturbance, or altering nutrient cycles [6,8–10]. Passengers can survive disturbances but are not catalysts of major changes in ecosystem biodiversity or function [7,11,12]. We propose to integrate driver and passenger species dynamics into the filter framework (Figure 1). In this adaptation, non-native passenger and driver species both pass through the filters and establish at which point passengers exist in the community with minimal broader impacts, while drivers alter the filters, thereby affecting the establishment and survival of other species. However, a challenge to understanding the role of an invasive species and its broader effects that will allow for the development of sustainable management protocols is determining the mechanisms through which it affects these filters [13].

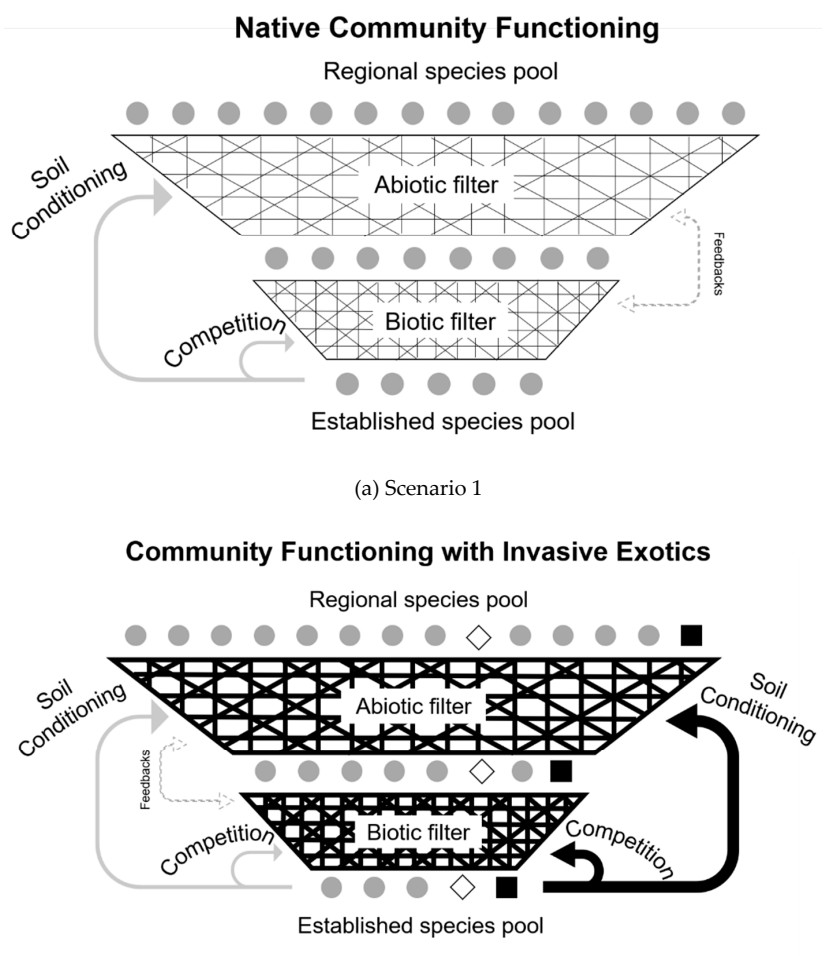

(a) Scenario 1

(b) Scenario 2

**Figure 1.** Proposed model combining the filter model with the passenger-driver model. Species from the regional species pool must pass through abiotic and biotic filters to become part of the established pool of species. The funnel shape indicates that not all species are able to pass through the filters. (**a**) Scenario 1 assumes native community functioning, or community dynamics of native plants (grey circles, e.g., *Lespedeza capitata*), while (**b**) Scenario 2 assumes community functioning with invasive exotic species present, or community dynamics when driver species are present. In Scenario 2, non-native passenger species = white diamonds and invasive driver species = black squares (e.g., *Lespedeza cuneata*) are able to pass through the filters and establish along with some native species = grey circles. The passenger has no additional major impacts while the driver species interacts with the abiotic and biotic filters (indicated by the black arrows), thereby affecting which species are able to pass through the filters to become part of the established community (indicated by the thickening of filter lines). The width of lines corresponds to the strength of impact: grey arrows are native impacts while black arrows are driver impacts. Dashed arrows indicate normal feedbacks between the abiotic and biotic filters that occur as part of community assembly processes.

The overall objective of this study was to explore how the invasive species *Lespedeza cuneata* functions as a driver and a filter through aboveground and belowground processes to affect species composition in restored grasslands, in order to guide management practices. We combined a regional study (field surveys) with a small-scale study (greenhouse experiment) to connect invasive dynamics with their broader impacts [13]. We asked three specific questions: (1) Is there a difference between species and functional group abundance and composition in areas where *L. cuneata* is present or absent within infested grasslands? (2) How does *L. cuneata* dominate native vegetation? and (3) How do native plants in different functional groups respond to aboveground and belowground interactions with *L. cuneata*?

## 2. Materials and Methods

### 2.1. Study Species

*Lespedeza cuneata* (Dum. Cours.) G. Don (Sericea lespedeza), is an invasive non-native perennial legume introduced in the late 1800s to the United States from Asia for forage, soil erosion control, and reclamation of degraded land [14]. *Lespedeza cuneata* easily overcomes native vegetation, forming near-monocultures. This invasive species has greater flower and seed production and attracts a higher frequency of pollinator visits than native *Lespedeza* spp., and produces seeds that can persist in the seed bank for over 20 years [15]. *Lespedeza cuneata* benefits from disturbances such as fire and is resilient as an adult due to a deep taproot [16].

Because attempts to control *L. cuneata* via fire, herbicide, grazing, and biocontrol have met limited long-term success, they should be replaced by efforts to manage this species without necessarily attempting complete removal [17–21]. To properly manage *L. cuneata*, an understanding of its interactions with the surrounding flora and subsequent impacts on the communities it invades is required [22]. Thus, exploring this species' potential to act as a driver in the filter model through aboveground and belowground mechanisms can provide insight into how it functions in invaded communities.

Aboveground, competitive shading by *L. cuneata* due to its large leaf area and high aboveground biomass has been observed [14,23], as well as its resistance to herbivory due to high tannin levels [17,24]. Belowground, intraspecific positive plant—soil feedbacks (PSFs) have been observed. In a positive PSF, a plant alters the soil to benefit its own growth, either directly or indirectly, via detriment to the growth of other plants [25–27] affecting invasion success [28]. PSFs of *L. cuneata* have led to high biomass and nodulation rates [26,29,30], modified soil microbial communities [31], high symbiotic nitrogen fixation [32], and allelopathic effects [33–37]. However, studies of *L. cuneata* soil conditioning have been limited in that the specific mechanisms responsible for changes in soil composition, the effects of these changes on co-occurring native legumes other than *Lespedeza* spp., and differences among grass, forb, and legume functional groups have yet to be identified.

### 2.2. Field Surveys

To investigate the patterns and outcomes of invasion by *L. cuneata*, 300 plots were established across 15 *Lespedeza cuneata*-infested grassland restorations in southern Illinois, USA, each with its own management history (see Table S1 in the Supplementary Material). Sites ranged in age from 2 to 45 years since establishment. The status of *L. cuneata* infestation prior to restoration was unknown, but *L. cuneata* was not introduced into the restorations. Field surveys were conducted in both early and late summer 2016 to capture seasonal phenological changes in vegetation. Within each site, 20 one-m$^2$ quadrats were established in a stratified random design in reference to a transect [38] with half the plots located in *L. cuneata*-absent areas (hereafter referred to as LCA plots) and half located in areas where *L. cuneata* was present (hereafter referred to as LCP plots), based upon protocols by Eddy and Moore [39] and a pilot survey in 2015 (unpublished data). A power analysis conducted on the data

from the pilot study indicated that a sample size of n = 9 per treatment (LCA/LCP) yields a power >0.8. Thus, 10 plots per treatment were sampled (total n = 20) [40,41].

The species present in each quadrat were identified (nomenclature according to [42]) and their abundance based upon the estimated canopy cover of plants rooted in the quadrat, recorded according to the modified Daubenmire scale [43]. In the second round of surveys, the plots were relocated, and data were collected as in the first survey.

## 2.3. Greenhouse Experiment

The greenhouse experiment was conducted at Southern Illinois University, Carbondale, Illinois, to investigate the mechanism through which *L. cuneata* interacts with native species. Phase one of the experiment, the "conditioning phase", consisted of growing *L. cuneata* in soil to condition it; phase two, the "response phase", consisted of the experimental trials [44]. Soil was collected from the upper 25 cm of a 2 m$^2$ area of a non-infested prairie restoration at Crab Orchard National Wildlife Refuge (CONWR) in January 2016 (37°42.027′ N, 89°03.928′ W). The soil type was a combination of two Oxyaquic Fragiudalf Alfisols, Ava silt loam 2–5% slope, and Ava silty clay loam 5–10% slope, severely eroded [45]. The soil was passed through a 1.25 mm sieve and mixed with sand to create a 1:1 soil:sand mixture that was placed in 10.2 cm × 10.2 cm × 35.6 cm plastic treepots (3670.7 cm$^3$ volume) (Hummert International, Earth City, Mo) to mimic deep grassland soils and minimize the chance of plants becoming root-bound. Seeds of *L. cuneata* collected from CONWR in autumn 2015 were germinated in Petri dishes and transplanted as 2–3 week old seedlings into half of the pots (n = 108) containing the collected and processed soil:sand mixture at a rate of twelve seedlings per pot. The other half of the soil did not have plants growing in it and became the "unconditioned" soil type. All pots were watered on the same watering regime (once to twice daily) and kept under 16 h photoperiod lamps. Pots were blocked by replicate (n = 4) and arranged 2 replicates per bench.

Plants were grown for 10 weeks from February 12, 2016 through April 22, 2016, a length of time over which effects of *L. cuneata* on its soil have been observed [35–37] and similar to the conditioning phase for previous plant-soil feedback studies [25,46–50]. The average ambient air temperature was 27.6 °C (maximum 42.4 °C; minimum 15.9 °C), the average relative humidity was 25.2% (maximum 86.7%; minimum 5.9%), and the average PAR was 221.6 μM/m$^2$s (maximum 1795 μM/m$^2$s; minimum 0 μM/m$^2$s). After 10 weeks, the aboveground and belowground biomass was removed before each soil type was re-sieved (2mm) and mixed within each replicate [51]. The available nitrogen was extracted from the conditioned and unconditioned soil samples (n = 10 per soil type) within 24 h of collection. The samples were analyzed for inorganic nitrogen availability ($NO_3$ and $NH_4^+$) via 2N KCl extractable nitrogen tests and for the presence and abundance of soil microbial communities through phospholipid fatty acid analysis (PLFA), at the SIUC Core Facility for Ecological Analyses. Thirteen fatty acids were analyzed as markers for 6 different bacterial/fungal functional groups: C14:0, C16:0, and C18:0, for non-specific bacteria; i-C15:0, a-C15:0, i-C16:0, and i-C17:0, for gram positive bacteria; C16:1_9 and C17:0 Δ 9,10, for gram negative bacteria; C18:1_9 cis and C18:2_9,12, for saprophytic fungi; C16:1_11 cis, for arbuscular mycorrhizae fungi; and 10 Me C16:0, for actinomycetes.

The experimental trials (phase 2) were designed to investigate the effect of soil conditioning and/or aboveground competition with *L. cuneata* on three functional groups. The experiment consisted of 2 soil treatments (conditioned or unconditioned soil), 3 competition treatments (no competition control, competition with native *Lespedeza capitata*, or competition with invasive *L. cuneata*), and 3 native target species from 3 functional groups (grasses, forbs, and legumes) with 4 replicates per treatment combination. Target grasses included *Andropogon gerardii* Vitman, *Sorghastrum nutans* (L.) Nash, and *Panicum virgatum* L.; target perennial forbs included *Solidago canadensis* L., *Penstemon digitalis* Nutt. ex Sims, and *Ratibida pinnata* (Vent.) Barnhart; and target legumes included *Chamaecrista fasciculata* (Michx.), *Desmodium* spp. (Torr.) Torr. & A. Gray, and *Senna hebecarpa* (Fernald) Irwin & Barneby. Seed unable to be collected from the field was supplemented with seed collected from the southern Illinois region in previous years (*Panicum virgatum* and *Ratibida pinnata*) or with seed purchased from

Prairie Moon Nursery, Winona, MN (*Chamaecrista fasciculata* and *Lespedeza capitata*). These target species, including the congener *L. capitata,* were all abundant native prairie species across the range of surveyed field sites that co-occurred with *L. cuneata*. The treatment with *L. capitata,* a native perennial legume closely related to *L. cuneata*, was included to differentiate between the general effects of competition and the specific effects of competition with the invasive *L. cuneata* on the native plants.

Seeds were germinated and transferred as 1–4 week old seedlings (depending on the growth rate of the species) into the soil in the greenhouse pots. Within the first 2 weeks, the plants were thinned to the most robust single seedling per species per pot. The plants were watered daily. Phase 2 ran 10 weeks, from 6 May to 17 July 2016. On 19 May 2016, all pots were moved to the Tree Improvement Center, on the SIUC campus, due to better temperature control. The average ambient air temperature was 26.3 °C (maximum 53.2 °C; minimum 11 °C), excluding data from 22 through 31 May 2016, due to a data logger malfunction. The plants ranged in maturity at the conclusion of the experiment.

Height and leaf number were measured weekly, while the final height, leaf number, root length, number of bacterial root nodules (present on legumes and visible with naked eye), and leaf chlorophyll, a proxy for nitrogen levels, quantified using a leaf chlorophyll meter [52,53], were measured at the conclusion of the experiment. The aboveground and belowground biomass of each plant was collected, oven-dried at 60 °C, and weighed. The specific leaf area was calculated by averaging the area of a leaf (cm) (measured using a Li-COR LI-3000A Portable Area Meter, LiCOR Biosciences, Lincoln, NB, USA) divided by its dry weight (g) for 3 leaves per plant. Soil pH and conductivity were measured using pH and conductivity meters according to protocol by the manufacturer (Fisher Scientific, Hampton, NH, USA).

## 2.4. Statistical Analysis of the Field Surveys

Percent cover data was transformed to the midpoint of each abundance class, *L. cuneata* abundance was excluded, and the data were standardized to the site unit total. Four diversity indices (richness, evenness, Shannon's H', Simpson's) and the average abundance of each functional group and origin group were calculated and compared between LCA and LCP plots with a two-way repeated measures mixed model, with the presence of *L. cuneata* nested within the site.

To investigate regional trends in *L. cuneata* invasion, data were averaged for each set of 10 plots per LCA/LCP treatment per site per survey (i.e., n = 15 LCA plots and n = 15 LCP plots per survey). Nonmetric multidimensional scaling (NMDS) was performed using Bray–Curtis dissimilarity values followed by vector fitting of 17 plots and 6 abiotic/management variables (Table 1). Species centroid plots were generated using the weighted averages of the 50 most abundant species. Repeated measures permutational analyses of variance (PERMANOVA) were run to test the significance of (1) *L. cuneata* presence and (2) site on LCA/LCP groups. Homogeneity of dispersion tests (PERMDISP) were performed to test the variability in the size of the LCA/LCP and site groups in ordination space.

Site-by-site analyses were conducted to investigate small-scale site-specific patterns not apparent in the regional analysis. The 10 LCA and 10 LCP plots within each site served as the replicates (i.e., n = 20 per site per round). NMDS and vector fitting analysis were performed separately for each site in the manner described above (excluding the abiotic/management vectors). Blocked repeated measures analysis of similarity (ANOSIM) tests were conducted to test for an effect of *L. cuneata*'s presence on species composition. PERMDISP was performed to test the variability of the dispersion of LCA and LCP plot groups in ordination space.

All data analyses were conducted in DECODA, PRIMER6, and SAS 9.4 [54–56]. Significance was accepted at $\alpha = 0.05$.

## 2.5. Statistical Analysis of the Greenhouse Experiment

The nitrate/nitrite levels, ammonium levels, and biomass of six soil microbial functional groups were compared between conditioned and unconditioned soil using pairwise t-tests (or rank tests for non-normal data). Mixed models were run to test for effects and interactions between 4 factors on the

growth of the 9 native target species: soil conditioning (fixed), competition (fixed), functional group (fixed), and replicate (random), using the measured dependent variables. Time was added as a fixed factor to this model to analyze repeated measures of height and leaf number data of both competitor species and target species over 8 weeks. Tukey HSD post-hoc tests were conducted to examine pairwise relationships. All data were log+1 transformed prior to analysis. Data from 206 plants were included in the analyses, giving a sample size of n = 206 (instead of the original 216) because 6 plants died and 4 pots of *A. gerardii* actually contained *Setaria viridis*. Two pots intended to contain *P. digitalis* that actually contained *S. canadensis* were analyzed as such because it was one of the study species. The same mixed model, repeated measures, and Tukey's analyses were conducted on the traits of the 2 competitor species, *L. cuneata* and *L. capitata*, using their measured dependent variables.

Modifications of Perkins and Nowak's [57] competition effect (CE) equation:

$$CE = (Biomass_{L.\ cuneata} - Biomass_{L.\ capitata})/(Biomass_{L.\ cuneata} + Biomass_{L.\ capitata}), \quad (1)$$

and Chiuffo et al.'s [25] PSF equation:

$$PSF = \ln(Biomass_{conditioned\ soil}/Biomass_{unconditioned\ soil}), \quad (2)$$

were used to quantify the competition effects and plant—soil feedbacks of the target species. Both indices were calculated using total, aboveground, and belowground biomass per each pair of target species from each treatment within each replicate, and then those values were averaged across replicates per species. The indices range from −1 to 1; positive values indicate that plants produced greater biomass when growing in conditioned soil or in competition, while negative values indicate that plants produced less biomass in conditioned soil or when in competition [57]. Indices were compared to zero using pairwise t-tests to indicate if a competition effect or PSF effect was occurring. All analyses were conducted in SAS 9.4 except for the pairwise tests, which were conducted in ggplot2 in R; Graphing was performed in SigmaPlot 11.0 (Systat Software, Inc. 2008, San Jose, CA, USA).

## 3. Results

### 3.1. Field Surveys

Across all plots, 275 species were identified (in addition to *L. cuneata*). LCP plots contained $9.8 \pm 0.6$ species per $m^2$, while LCA plots contained $10.2 \pm 0.8$ species per $m^2$ ($t_{28} = 0.42$, $p = 0.68$). The total plant cover excluding *L. cuneata* was 20.4% higher in LCA compared to LCP plots (mean ± se; LCA plots = $103.8 \pm 1.6\%$, LCP plots = $83.4 \pm 1.9\%$), with an *L. cuneata* cover of $20.7 \pm 1.4\%$ in LCP plots. The cover of two functional groups was greater in LCA than LCP plots: grasses (LCA plot mean = $38.2 \pm$ se $6.4\%$, LCP plots = $18.4 \pm 3.0\%$, $F_{15,132} = 4.14$, $p < 0.0001$) and legumes (LCA plots = $14.2 \pm 2.7\%$, LCP plots = $7.4 \pm 1.4\%$, $F_{15,152} = 4.2$, $p < 0.0001$).

Regionally, community composition did not differ between LCA and LCP plots (PERMANOVA pseudo-$F_{1,59} = 0.24$, $p = 1.0$). There were significant differences among sites (PERMANOVA pseudo-$F_{14,59} = 4.69$, $p = 0.001$), between LCA and LCP plots nested within site (PERMANOVA pseudo-$F_{15,59} = 5.41$, $p = 0.001$), and between surveys per site (PERMANOVA pseudo-$F_{15,59} = 3.26$, $p = 0.001$). Sixteen of the 23 tested vectors were significantly related to the NMDS ordination (Table 1, Figure 2a). The abundance of invasives and natives and the fire history vectors had the strongest correlations (indicated by Max R values). The species centroid plots indicated a positive association between fire/herbicide management and native grasses and a negative association between the management vectors and non-native grasses (Figure 2b, Table S2 in the Supplementary Material). Tests of the dispersion of plots in the regional ordination illustrated the variability in the dispersion of points for each site (indicated by a significant PERMDISP between site groups) but no differences in the variability of LCA and LCP plot dispersion.

**Table 1.** The 23 vectors fitted to the field survey nonmetric multidimensional scaling (NMDS) ordinations, grouped by category (table columns). Vectors for functional and origin groups are based on abundance data; abiotic/management vectors were only included in the regional analysis. Max R and *p* values indicate the vector fit to the regional analysis (Figure 2a).

| Functional Groups | Origin Groups | Diversity Measures | Survey Variables | Abiotic/Management |
|---|---|---|---|---|
| Grass (R = 0.36, *p* = 0.02) | Native (R = 0.84, *p* < 0.0001) | Shannon diversity (R = 0.44, *p* = 0.002) | Site (R = 0.26, *p* = 0.15) | Site age (2–45 years) (R = 0.19, *p* = 0.34) |
| Forb (R = 0.59, *p* < 0.0001) | Non-Native (R = 0.43, *p* = 0.008) | Simpson diversity (R = 0.47, *p* = 0.001) | Survey round (R = 0.04, *p* = 0.96) | Site size (0.4-14.16 ha) (R = 0.37, *p* = 0.018) |
| Legume (R = 0.32, *p* = 0.05) | Invasive (R = 0.86, *p* < 0.0001) | Evenness (R = 0.41, *p* = 0.006) | *L. cuneata* presence (R = 0.06, *p* = 0.90) | History of fire (yes, no) (R = 0.74, *p* < 0.0001) |
| Sedge/rush (R = 0.13, *p* = 0.60) | | Richness (R = 0.43, *p* = 0.003) | *L. cuneata* abundance (R = 0.23, *p* = 0.23) | *L. cuneata* herbicide treatment (yes, no) (R = 0.51, *p* = 0.0003) |
| Vine (R = 0.32, *p* = 0.04) | | | | Soil type (Alfisols, Inceptisols, Enstisols) (R = 0.53, *p* < 0.0001) |
| Woody (R = 0.13, *p* = 0.58) | | | | Slope (0–20 degrees) (R = 0.59, *p* = 0.0002) |

On an intra-site scale, eight of the 15 sites had significantly different species composition between LCA and LCP plots, with the presence of *L. cuneata* (the plot selection criterion) and its abundance driving this separation (Table 2). Four of the eight sites had higher forb abundance in LCP compared with LCA plots, while three sites had higher grass abundance in LCA compared with LCP plots. The significant difference in composition between LCA and LCP plots in six of the eight sites was supported by a nonsignificant difference in dispersion, which indicates that there is no significant variability in the size of the LCA and LCP groups, and thus the detection of a significant ANOSIM is based on differences in community composition. The significant PERMDISP result for the other two sites, however, indicates that the variability in size of the LCA and LCP groups could be contributing to the difference found between the plot types, rather than their separation being driven by differences in species composition alone (Table S3 in the Supplementary Material).

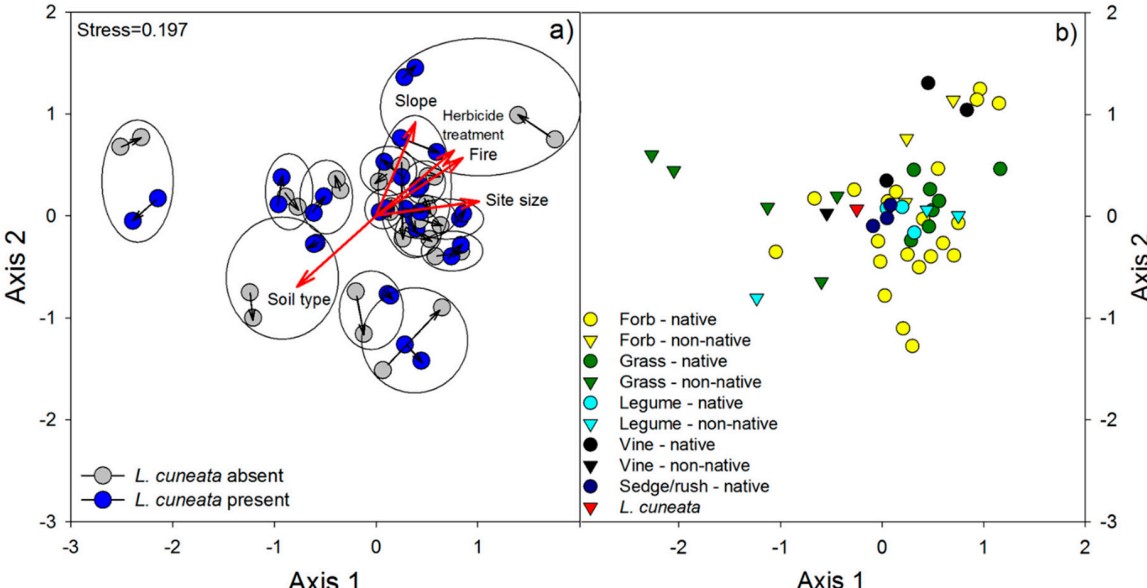

**Figure 2.** Panel (**a**) NMDS ordination (stress value = 0.197) comparing the grassland community composition of *L. cuneata*-absent (LCA) and *L. cuneata*-present (LCP) plots of all sites on the regional scale. Each point represents the average community composition of all 10 LCA or LCP plots per site per survey. Points from the first and second survey for each plot are connected with black arrows to illustrate how the plots changed over time. Pairs of LCA and LCP plots from the same site are within ellipses to clarify interpretation using the ellipse tool in SigmaPlot. Five abiotic vectors (red lines) were fitted to the ordination significant at $\alpha < 0.05$ (Table 1). Panel (**b**) Functional group centroids of the NMDS ordination (Panel a) calculated from the weighted averages of the 50 most abundant species in the field surveys.

**Table 2.** Effect of the presence of *L. cuneata* on community composition. Repeated measures analysis of similarity (ANOSIMs) with LCA/LCP plots as groups and survey as blocks for each site. Max R = ANOSIM test statistic. Significance of *p*-values accepted at $\alpha = 0.05$ and shown in bold.

| Site | Max R | *p*-Value |
|---|---|---|
| Bass Pond East (CONWR) | 0.06 | 0.02 |
| Bass Pond West (CONWR) | 0.03 | 0.09 |
| Cache River | 0.09 | 0.009 |
| Cave Creek Glade | 0.13 | 0.0007 |
| Dixon Springs | 0.07 | 0.04 |
| Faulkner-Franke Prairie | 0.05 | 0.02 |
| Hampton Complex 6 (CONWR) | −0.04 | 0.77 |
| Headquarters Prairie (CONWR) | 0.16 | <0.00001 |
| North Prairie 1 (CONWR) | 0.08 | 0.0005 |

**Table 2.** *Cont.*

| Site | Max R | *p*-Value |
|---|---|---|
| North Prairie 2 | 0.01 | 0.26 |
| Pennant Bar 1 | 0.05 | 0.06 |
| Pennant Bar 2 | 0.03 | 0.11 |
| Postage Stamp (CONWR) | 0.02 | 0.15 |
| Pyramid State Park Denmark | 0.05 | 0.06 |
| Pyramid State Park Galum | 0.19 | <0.0001 |

### 3.2. Greenhouse Experiment

Nitrate/nitrite levels were greater in unconditioned compared to conditioned soil, while ammonium levels were not significantly different (mean ± se): nitrate/nitrite (unconditioned = $8e^{-3} \pm 6e^{-4}$ mg/g soil, conditioned = $7e^{-4} \pm 6e^{-5}$ mg/g soil, $t_{9.2} = 11.5$, $p < 0.001$) and ammonium (unconditioned = $9.4e^{-4} \pm 3.56e^{-4}$ mg/g soil, conditioned = $3e^{-4} \pm 6e^{-5}$ mg/g soil, Mann–Whitney U = 43, $T_{10,10} = 98$, $p = 0.6$). There was a higher abundance of two fungal communities in conditioned soil (mean ± se): saprophytic fungi (unconditioned = $1.2 \pm 0.10$ nmol/g soil, conditioned = $1.82 \pm 0.25$ nmol/g soil, $t_{11.8} = 2.3$, $p = 0.04$) and arbuscular mycorrhizal fungi (unconditioned = $1.04 \pm 0.07$ nmol/g soil, conditioned = $1.38 \pm 0.11$ nmol/g soil, $t_{15} = 2.6$, $p = 0.02$). Conditioned soil produced target plants with greater belowground biomass and higher leaf chlorophyll levels than unconditioned soil (Figure 3). Competition (with both *Lespedeza* competitors) reduced total and belowground biomass of all target plants and root length of forbs and legumes (Figure 4). The functional group by time interaction on the repeated measures height data indicated that the functional groups had different growth rates, with grasses growing at a faster rate than forbs and legumes (Figure 5).

Soil conditioning and competition with *L. cuneata* affected grass traits in a positive manner (excluding specific leaf area). Competition with *L. cuneata* negatively affected forbs and legumes for most traits, although the leaf chlorophyll of legumes and forbs was increased by soil conditioning (legumes) or competition with *L. cuneata* (forbs). When competing with *L. cuneata*, the grass *Andropogon gerardii* experienced a positive PSF effect on total, aboveground, and belowground biomass while another grass, *Sorghastrum nutans*, experienced a positive PSF on belowground biomass (Figure S1 in the Supplementary Material). One forb, *Ratibida pinnata*, experienced a positive PSF on belowground biomass in the absence of competition. In unconditioned soil, the total and belowground biomass of the forb and legume functional groups (as well as the aboveground biomass of the legumes) experienced a significant negative competition effect. In conditioned soil, the total and belowground biomass of the forbs was more negatively affected by competition with *L. cuneata* compared to *L. capitata*.

Soil conditioning did not affect the traits of either *Lespedeza* competitor species (all traits, $p > 0.05$). However, there was a significant effect of the target species' functional group on the growth of the competitor species for 9 of the 11 measured traits, with the Tukey's tests indicating different responses of the invasive competitor to competition with different functional groups, but no difference in responses of the native competitor (Figure 6). When grown with forbs, the biomass, final plant height, and leaf number of *L. cuneata* individuals were greater than when in competition with the other two functional groups (Figure 6). Competition with legumes resulted in the shortest roots, smallest number of root nodules, and lowest leaf chlorophyll levels in *L. cuneata* individuals. The repeated measures analysis on height and leaf number revealed a significant interaction between competitor species, functional group, and time on the height and leaf number of competitor species, with the competition with forbs producing the tallest *L. cuneata* individuals, with the most leaves, and the competition with legumes the shortest, with the least leaves. The competition with all functional groups produced similar sized *L. capitata* individuals with a similar number of leaves (Figure 7).

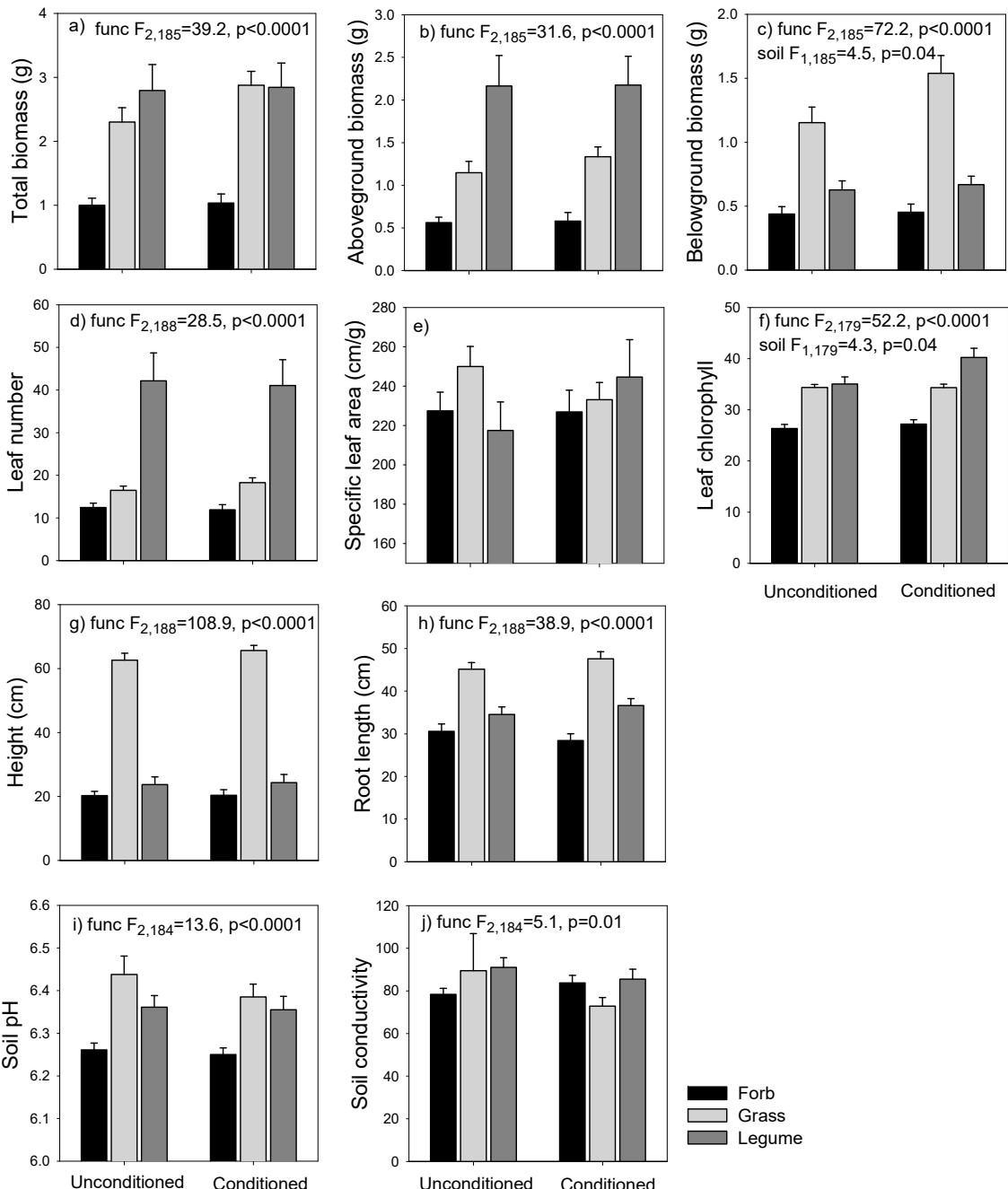

**Figure 3.** The effect of conditioned and unconditioned soil on the growth of target plants from three functional groups: forbs, grasses, and legumes. Ten traits were measured at the conclusion of the greenhouse experiment (mean ± se): (**a**) total biomass, (**b**) aboveground biomass, (**c**) belowground biomass, (**d**) height, (**e**) root length, (**f**) leaf number, (**g**) specific leaf area, (**h**) leaf chlorophyll levels, (**i**) soil pH, and (**j**) soil conductivity. Total n = 206. Replicates per bar = 36 except for unconditioned soil × forbs, where n = 33, unconditioned soil × grasses, where n = 33, conditioned soil × forbs, where n = 33, and conditioned soil × grasses, where n = 35. The F statistic and *p*-values included on the graphs correspond to either a significant soil or functional group factor ($\alpha < 0.05$). There were no significant interactions between soil and functional group treatments.

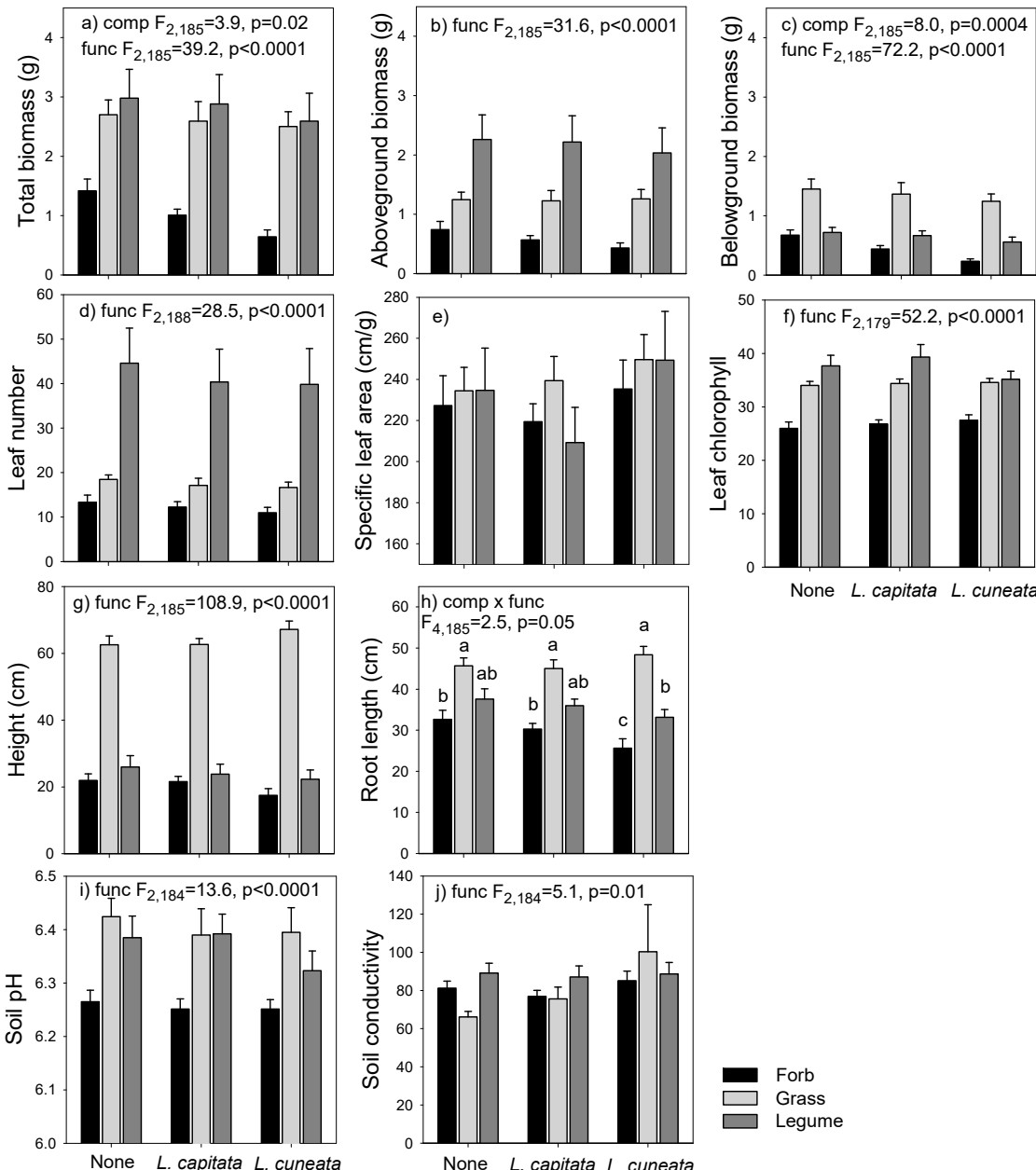

**Figure 4.** The effect of three competition treatments (none, *L. capitata*, and *L. cuneata* competition) on the growth of target plants from three functional groups: forbs, grasses, and legumes. Ten traits were measured at the conclusion of the greenhouse experiment (mean ± se): (**a**) total biomass, (**b**) aboveground biomass, (**c**) belowground biomass, (**d**) leaf number, (**e**) specific leaf area, (**f**) leaf chlorophyll levels, (**g**) height, (**h**) root legnth, (**i**) soil pH, and (**j**) soil conductivity. Total n = 206. Replicates per bar = 24, except for no competition × forbs, where n = 21, no competition × grasses, where n = 22, *L. capitata* × forbs, where n = 23, *L. capitata* × grasses, where n = 23, *L. cuneata* × forbs, where n = 22, and *L. cuneata* × grasses = 23. The F statistic and *p*-values included on the graphs correspond to the significant competition, functional group, or competition by functional group factor (α < 0.05). Mean values of bars sharing the same letter are not significantly different for the competition by functional group interaction on root length (interaction effects on other variables were not significant).

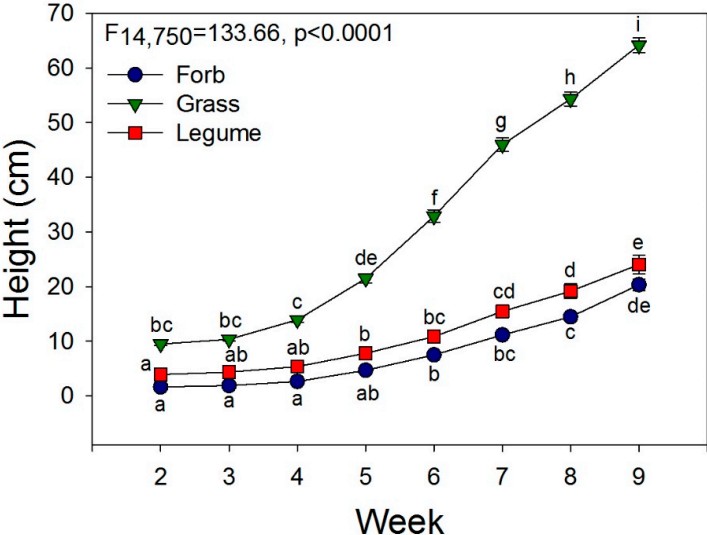

**Figure 5.** The effect of the interaction between functional group and time ($F_{14,750}$ = 133.66, $p < 0.0001$) on the height (mean ± se) of three functional groups. Different letters indicate significant differences. Total n = 206. Mean values of bars sharing the same letter are not signficiantly different ($p < 0.05$). Blue circle = forb, green triangle = grass, red square = legume.

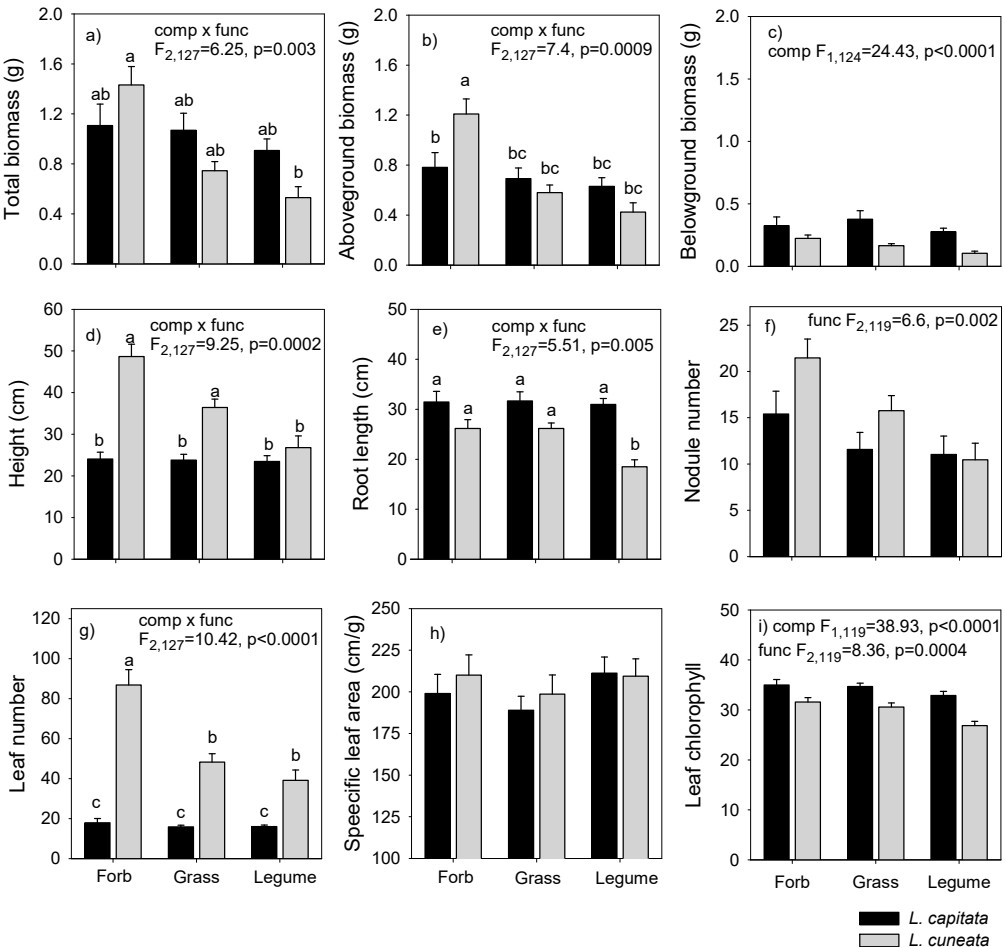

**Figure 6.** The effect of competition with three different functional groups on the growth of the competitors, native *L. capitata* and invasive *L. cuneata*. Nine traits were measured at the conclusion of the 9-week greenhouse experiment (mean ± se): (**a**) total biomass, (**b**) aboveground biomass,

(**c**) belowground biomass, (**d**) height, (**e**) root length, (**f**) number of root nodules, (**g**) leaf number, (**h**) specific leaf area, and (**i**) leaf chlorophyll levels. Total n = 139. Replicates per bar = 24, except for forb × *L. capitata*, where n = 23, grass × *L. capitata*, where n = 23, forb × *L. cuneata*, where n = 22, and grass × *L. cuneata*, where n = 23. The F statistic and *p*-values included on the graphs correspond to a significant functional group, competitor species, or functional group by competitor species factor on competitor species' growth. Mean values of bars sharing the same letter are not signficiantly different (α < 0.05) for the interaction between functional group and competitor species treatments on the total and aboveground biomass, root length, height, and leaf number (interaction effects on other variables were not significant).

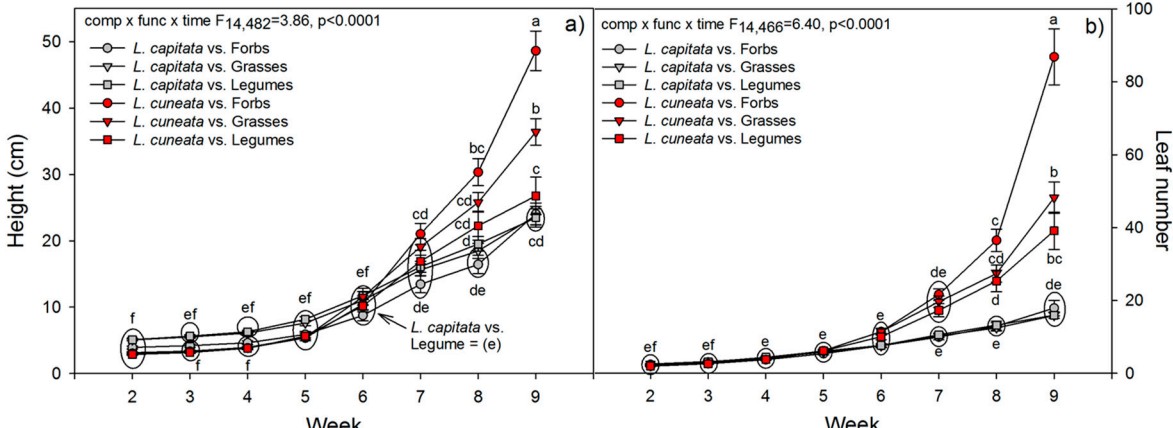

**Figure 7.** The effect of the interaction between competitor species, functional group, and time on the repeated measures height (panel a, $F_{14,482} = 3.86$, $p < 0.0001$) and leaf number (panel b, $F_{14,466} = 6.40$, $p < 0.0001$) of the competitor *Lespedeza* spp. over the course of the greenhouse experiment. Total n = 206. Mean values of bars sharing the same letter are not signficantly different ($p < 0.05$). Points grouped by circles share the same letter. Grey = *L. capitata*; red = *L. cuneata*; circles = forbs; triangles = grasses; squares = legumes.

## 4. Discussion

Grassland restoration and management should incorporate the likelihood that exotic species will spread under climate change [58], and recognize that adaptive management practices will be necessary to create sustainable grassland restorations [59]. Our results reinforce the need to understand the mechanisms through which exotic species such as *Lespedeza cuneata* compete with native species in restorations to better sustain the integrity and ecosystem services that grasslands provide [60,61].

### 4.1. Patterns of Invasion: Field Surveys

The community composition of invaded and uninvaded *L. cuneata* patches within infested sites differed at some sites, but not regionally, indicating a scale effect [13]. Moreover, the widespread occurrence of *L. cuneata* across the landscape suggests that dispersal was not a limitation for invasion. As the presence of *L. cuneata* was unrelated to community composition regionally, its effects were likely being driven by differences between sites such as site quality; site size, which could be an indication of greater heterogeneity in larger sites supporting greater species richness [62]; differences in species diversity, indicating that grasslands vary in species composition regardless of the interactions with invasive species [63]; soil types, which varied across three soil orders; differences in the abundance of native, non-native, and invasive species; and differences in degree and type of management [13,64]. It is likely that multiple factors are interacting to determine species composition, as observed in California grasslands where the abundance of invasive species, the grazing regime, and the soil type interact to determine the abundance of native species [65].

By contrast, differences between invaded and uninvaded patches were observed in half the surveyed sites when analyzed within-sites. In some sites, the presence of *L. cuneata* coincided with

differences in the abundance of functional groups and community composition, while in others, the characteristics of each site and its management appeared to play a stronger role in determining species composition. Thus, *L. cuneata*'s relationship to the vegetation in the sites that it invades is likely dynamic, scale-dependent, and related to the functional group and phylogenetic similarity of established native species [21].

At the site scale, this study suggests that invasion success is related to the functional group identity of both the invader and the established plants. The observed higher abundance of legumes and grasses in uninvaded patches than invaded patches suggests either that *L. cuneata* is a superior competitor and suppresses the growth of these functional groups where it occurs, or that it invades where these functional groups are already at low abundance. Prior studies support the explanation that these functional groups may offer greater resistance to *L. cuneata*'s invasion than other functional groups, such as non-leguminous herbaceous plants. Grassland plants are most successful at resisting the invasion of species belonging to their own functional group, due to limiting similarity [66–69]. Because the competition between species of the same functional group is expected to be more intense due to their overlapping niches, limiting similarity predicts that an invasive plant will prevent the further establishment of native plants in the same functional group [70,71]. Fargione et al. [67] observed that in grasslands, grasses prevent the establishment of invasive species (regardless of the functional group of the invader) to a greater degree than other groups. This notion supports the idea that the poor establishment of *L. cuneata* in areas with high grass abundance is due to a weak ability to compete with grasses [4], and the observations of relatively slow seedling emergence and resprouting from the perennial root system in the spring [16,72]. While legumes were still present in LCP plots, their overall abundance was half that observed in LCA plots, which supports prior findings of limiting similarity of legume invaders that had a limited ability to invade legume monocultures [73]. We observed a significantly greater abundance of grass and legume species in uninvaded patches, suggesting that, on a local scale, grasses can suppress an invasive legume, and native legumes can exclude it. Our findings align with observations of the lower success of *L. cuneata* in restored plots with higher abundances of $C_4$ grasses and legume species, compared to non-restored plots with lower abundances of these functional groups [74].

*4.2. Mechanisms of Invasion: Greenhouse Experiment*

Changes to soil properties due to the growth of *L. cuneata*, the first step in a plant–soil feedback [75], were observed after 10 weeks of conditioning. A significantly greater fungal biomass in conditioned than unconditioned soil is supportive of speculations that *L. cuneata* modifies its soil via changes to soil microbial communities [30,31], which in turn can affect its competitiveness against native species [76]. However, because most plant species, including *L. cuneata* [77], have an arbuscular mycorrhizae association, finding elevated fungal biomass in conditioned soil should not be surprising [78].

Our expectations of greater levels of nitrogen in conditioned compared with unconditioned soil were not fulfilled. However, our results are not unprecedented, because comparisons of nitrogen levels in *L. cuneata* conditioned and unconditioned soil have found greater levels in conditioned soil [30], lower levels in invaded soil [79], and equal amounts between the two soils [26]. In our study, microbial immobilization of nitrogen in response to plant carbon inputs in the conditioned soil is a likely explanation of lower nitrate levels. As the roots of *L. cuneata* individuals grew, they likely provided an increasing input of carbon, which would have been broken down by microbes [80]. Because this microbial activity requires a nitrogen source, it is likely that the active soil's microbial community depleted the total available nitrogen in the conditioned soil [80].

This study supports the interpretation that aboveground competition plays a larger role than belowground effects in determining which plants can coexist with *L. cuneata* [30]. The PERMANOVA results exploring which greenhouse experiment treatments drove the overall differences between target plants found a nonsignificant effect of soil (supporting the conclusion that the soil did not affect plants across all measured traits), but a significant effect of competition. Studies of the relative strength of

PSFs and competition find the latter to be a stronger influence and/or to alter soil relationships [81,82]. Soil conditioning only benefited the growth of grasses, which supports previous findings showing a greater biomass of the native grass *S. nutans* when grown in *L. cuneata*-invaded than in uninvaded soil [26]. In this case, changes to the soil could be enhancing the competitive ability of the natives by reducing the amount of native pathogens or promoting the growth of beneficial microbes, including mycorrhizae, to natives [81]. While PSFs may play a role in determining the invasiveness of plants, it appears that for *L. cuneata*, soil effects are minimized in comparison to competition, even under greenhouse conditions, where PSFs are likely to be most evident [83].

The growth of *Lespedeza cuneata* did not differ between the two soil conditioning treatments, contrary to direct positive plant–soil feedbacks observed between many invasive species and their soil [84], suggesting that *L. cuneata* experiences an indirect PSF through its effects on neighbor species rather than directly benefiting its own growth [25,27]. Because *L. cuneata* plants were not grown individually in both soils, the PSFs of the *L. cuneata* individuals could not be calculated. Considering the specific effects of competition on measured traits (mixed models) rather than whole plant growth can provide a detailed picture of how an invasive species responds to competition with different functional groups. Comparing the individual functional traits of the *Lespedeza* competitors in response to growing with three target functional groups revealed only one *L. capitata* trait affected by functional group but 9 of the 11 *L. cuneata* traits differing depending upon the functional group with which it was competing. Competition with forbs produced the best performing *L. cuneata* individuals, while competition with legumes produced the poorest performers. The difference in responses between the native and non-native *Lespedeza* could be due to their origin (native vs. non-native), the greater leaf production of *L. cuneata*, different growth forms (branched, bush-like *L. cuneata* vs. slender *L. capitata*), or different bacterial associations [29,85]. These results illustrate that the native *Lespedeza capitata* is more able to coexist with species across functional groups than the non-native *L. cuneata*.

### 4.3. Role as a Driver and Filter

This study supports the role of *L. cuneata* as a driver species, because, through its competition with other species and ability to condition the soil, it leaves a lasting effect on the ecosystem it invades [6]. These mechanisms alter current filters, with soil conditioning altering the nutrient and microbial community abundance (i.e., abiotic filters) and the invasive species exerting stronger competitive effects than a native competitor (i.e., a biotic filter) [86–88]. Because *L. cuneata* altered these filters in ways that benefit its growth and suppress that of other native species, it can be considered a driver [89].

The ability of *L. cuneata* itself to act as a filter was not supported in this study. The field surveys suggested that established native grasses and legumes act as superior competitors and may locally filter out *L. cuneata*. This observation supports the idea that the most competitively dominant species in an area, either an invasive species or an already established native, will filter the establishment of additional species [90]. In the greenhouse experiment, most functional traits of grasses were either unaffected or increased when in competition with *L. cuneata*, while competition with grasses reduced *L. cuneata*'s growth in several traits. Legume growth was only slightly reduced under competition with *L. cuneata*, while among the functional groups, legumes suppressed the growth of *L. cuneata* to the greatest degree in the majority of traits. Because these results suggest that competition between grasses/legumes and *L. cuneata* results in the facilitation of native grasses [91], limiting similarity of other legumes, and detrimental effects to *L. cuneata*, these functional groups act in order to filter the establishment of *L. cuneata*.

### 4.4. Management Implications and Outlook

These findings reaffirm the prioritization of managing to control *L. cuneata* in the grasslands that it invades, in order to sustain ecosystem services [20,21]. Combining the observational and experimental findings results in several broad management recommendations. First, managing high grass abundance will help buffer a grassland from *L. cuneata*'s invasion or its further spread. Second,

increasing diversity by planting legumes, rather than non-leguminous forbs, can help increase the resistance to invasion and the quality of a grassland [92,93]. To promote legume growth, inoculating with rhizobia should be considered, as well as options for reducing nutrient limitation [94,95]. Finally, sites should be assessed and monitored on a local scale to determine the specific impacts of *L. cuneata* on the functional groups of individual grasslands, because this species does not operate in a consistent manner. Including these recommendations in a larger invasive plant management framework can increase the chances of achieving a sustainable grassland restoration [96,97].

## 5. Conclusions

*Lespedeza cuneata* acts as a driver in the ecosystems it invades by altering the abiotic and biotic filters to affect native species on a functional group basis, providing support for a combined filter-driver framework as a useful tool for studying invasive impacts (Figure 1). The variation in the impacts of *L. cuneata* on functional groups suggests that this invasive species may not be dominantly suppressive of all members of the communities it invades. The contingency of site-specific relationships apparent in the field survey, coupled with the controlled greenhouse experiment, does not support the role of this invasive species as a strong filter, providing hope that the occurrence of *L. cuneata* does not determine community dynamics and that some native species have the potential to act as strong filters. Future studies should consider the interspecific competition between *L. cuneata* and native dominants, especially grasses and legumes, during the invasion process.

**Supplementary Materials:** The following are available online at http://www.mdpi.com/2071-1050/12/15/5951/s1. Figure S1: The plant-soil feedback responses (on total (a), above- (b), and belowground (c) biomass) of the nine target species (codes refer to the first letter of the genus and specific epithet, respectively [Table S2]) to soil conditioning without competition (panel a) and in the presence of competition with *L. cuneata* (panel b) and the competition effect responses (on total (a), above- (b), and belowground (c) biomass) on the nine target species, comparing the competition with *L. capitata* to the competition with *L. cuneata* in unconditioned (panel c) and conditioned soil (panel d). The PSF was calculated as $\ln(\text{Biomass}_{cond}/\text{Biomass}_{uncond})$ within each replicate and averaged per target species (mean ± se). Positive responses indicate that the target species produced greater biomass in conditioned soil, while negative responses indicate that they produced less biomass in conditioned soil. The competition effect was calculated as $(\text{Biomass}_{Lcun}-\text{Biomass}_{Lcap})/(\text{Biomass}_{Lcun}+\text{Biomass}_{Lcap})$ within each replicate and averaged per target species (mean ± se). Positive responses indicate that the target species produced greater biomass in competition with *L. cuneata* than in competition with *L. capitata*, while negative responses indicate that they produced less biomass in competition with *L. cuneata* than in competition with *L. capitata*. Significant PSFs and competition effects (compared to 0) are denoted with *. All calculations are averages of 3–4 replicates, excluding *A. gerardii*, *P. digitalis* and no-competition PSFs, which are averages of only 2 replicates due to misidentified and dead plants; Table S1: Field site background, including location, survey dates, and establishment and management history (Casey Bryan, Chris Evans, Scott Crist, Joe Nelson, personal communication, 2016); Table S2: Fifty most abundant species observed during the two field survey periods across all 15 sites. * Two-letter genus/species are used for target species codes in Figure S1 (Ds = *Desmodium* spp. and Sh = *Senna hebecarpa*); Table S3: Results of vector fitting for each of the 15 site ordinations. R and *p*-values are bold if significant at $\alpha < 0.05$.

**Author Contributions:** Conceptualization, E.M.G. and D.J.G.; methodology, E.M.G.; formal analysis, E.M.G.; investigation, E.M.G.; data curation, E.M.G.; writing—original draft preparation, E.M.G.; writing—review and editing, E.M.G. and D.J.G.; supervision, D.J.G.; funding acquisition, E.M.G. All authors have read and agreed to the published version of the manuscript.

**Funding:** This research was funded by the Illinois Association of Environmental Professionals.

**Acknowledgments:** Permission to conduct the field work was obtained from the U.S Fish and Wildlife Service, Illinois Department of Natural Resources, Illinois Nature Preserves Commission, University of Illinois Extension, and Shawnee National Forest. The greenhouse experiment was conducted in the Horticultural Research Center, SIUC.

**Conflicts of Interest:** The authors declare no conflict of interest. The funders had no role in the design of the study; in the collection, analyses, or interpretation of data; in the writing of the manuscript, or in the decision to publish the results.

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
