# Peer review of "Identifying Sustainable Grassland Management Approaches in Response to the Invasive Legume Lespedeza cuneata: A Functional Group Approach"

_sustainability, doi:10.3390/su12155951_

Round 1

Reviewer 1 Report

The study presented here has several objectives: Firstly, to investigate which type of vegetation can act as filter to prevent invasion of the nonnative legume Lespedeza cuneata into north American prairie; secondly, to investigate to which extent this species can act as a filter itself and alter the environment so that its own establishment is facilitated and that of co-occurring native species inhibited; thirdly  - although this is not always stated clearly -, to which extent these effects are stronger than those of its native congener Lespedeza capitata; and finally, to extract management recommendations from the results.

To this end, the authors draw on vegetations surveys from invaded and uninvaded areas in fifteen prairie restoration sites, and on a greenhouse experiment in which both lespedezas were grown singly or together with three native species each of grasses, forbs and legumes (“functional groups”) and in soil conditioned by previous growth of Lespedeza cuneata.

Before I start on my comments I would like to thank the authors for the opportunity to review this study; more importantly, I wish to commend the first author especially for the enormous effort that must have gone into field and greenhouse work, not to mention statistical analysis and writing.

Yet I have come to the conclusion that only after thorough revision will this manuscript fulfil the high expectations of the authors themselves, and of their readership. There are altogether too many variables under consideration, and too many graphs presented, that obstruct the view of those results that might truly help answer their questions.

The manuscript also seems to have been hastily compiled and shortened; this shows in the abstract already (“to” or “can” missing before “help” in first sentence; “whether” should be “if”, I think; competition was not between the two lespedezas but between each of these and the species, or at least so it reads in the manuscript). But those are minor details that should be judged by a native speaker.

The overall setting and objective

Although not all information available on the internet is peer-reviewed, it seems well established that Lespedeza cuneata, hereafter LC, a) benefits from site disturbance, b) flowers and produces seeds in much greater abundance that its native congeners but is comparatively weak in its juvenile stage and should therefore be headed off early in its establishment, c) derives its resilience as an adult, among others, from the formation of a deep strong taproot. I refer mostly to Cory Guckers literature compilation in the Fire Effects Information System.

Yet the authors do not a) and b) into account in their investigations and discussion. While it is a valid assumption that LC alters the soil to its benefit – how relevant might this be when, for example, prairie restoration sites are burnt regularly, often in an attempt to manage invasives, after which adult LC resprouts and numerous gaps allow banked LC seeds to germinate? The fact that it has not been investigated before does not mean it is relevant; perhaps quite to the contrary.

As for juvenile growth and taproot development, the authors do create a good setting for investigating just that but do not make thorough use of it.

The field survey

What information on site establishment and management was available to the authors is provided in the supplementary materials; it differs greatly in quality and the sites differ in age by decades. There is not even information for all sites on when LC became so problematic that it had to be controlled (or the attempt made). Moreover, several sites contain other problematic invasive species.

The authors are not to be blamed for this. But it is extremely difficult, in such a setting, to draw conclusions from standard vegetation surveys on whether a given site is resistant to invasion, or has just not yet been invaded, and which factors might contribute to resistance, and which are altered by LC itself. This shows in the presentation of the results and the discussion.

Fig. 2 is almost unreadable (and figure S1 is, I am afraid). Vectors of site development are mostly short and pointing apparently at random, so why present them? Variable vectors are a jumble of abiotic site conditions such as soil type, slope; of management - fire, herbicide; and of vegetation variables likely influenced by LC itself (diversity, evenness…). I suggest to concentrate on the first, and check which of these correlates best with axis 1.

Fig 3, left pane is obsolete; species codes overlap too much and anyway an outsider cannot read them without referring to the species list in the annex, which is cumbersome; the right pane, however, is important. It indicates a negative correlation of non-native grasses with axis 1 and a positive one with native abundance, but also non-native forbs and legumes. Fire and herbicides also have a positive correlation with this axis, see Fig. 2, so that they might to some extent suppress non-native grasses but not necessarily forbs and legumes.

And that is basically what I can draw from the vegetation survey in its presented form.

Why were two surveys conducted? Was this before and after management events or “just” early and late summer? Was life stage of LC on the sites perhaps noted, such as presence/cover of juveniles and adults? Total vegetation cover is high, but was open soil (germination niches) noted?

LC has unfortunately a history of artificial selection and seeding at a large scale in the U.S. Would the authors be able to state what kind of seed were used in site restoration – wild collections, or commercial seed? Grass cultivars? As some of the sites are so old: Were they seeded with natives at all? This might then be used as another truly independent variable.

If this is not feasible then I suggest drastically shortening this chapter, including the annex.

The greenhouse experiment

Phase 1, the soil conditioning, is described with detail and seems to have been done with great care. From phase 2, the competition x soil experiment, I miss this information:

Why was congener L. capitata chosen?

What was the seed source? Wild collections for the dicots (“Desmodium spec. …”), I suppose, but the grasses – also wild or cultivars?

Was phase 2 duration 10 weeks to prevent conditioning of the soil by LC (as happened in the 1st phase within that time?) Were plants still in their juvenile stage then, or mature – growing inflorescences?

Was the experiment watered? I suppose so, and if so, the authors thus took the edge off the presumed juvenile weakness of LC. All the more important to pay attention to initial growth rate, which might be derived from the data they have - and which might provide insight into which functional group, in the field, might have the most impact at this critical stage.

Treatment combinations were 2 soil conditions x 3 competition treatments x 9 species = 54; these were replicated four times each, too little to allow for errors and dieback and for robust statistical analyses even with contemporary tools. Consequently, replicates were pooled; but this is not stated clearly, nor is number of replicates given for the different graphs and statistical analyses. Moreover, results need to be interpreted very carefully.

For example, Fig. 4 shows a clear increase of belowground grass biomass on LC-conditioned soil: But Fig. S2c reveals that of the three grasses, only Sorghastrum nutans grew more root biomass in conditioned soil whether LC was present or not. Andropogon gerardii required both conditioned soil and LC presence in the pot to do so.

Fig. 4 and 5 reveal mostly no differences, or differences in measured plant traits only between functional groups - but the latter is to be expected in groups as different as grasses and dicots. I suggest keeping only those graphs that illustrate clear soil effects, competition effects and functional group x competition effects; supplementing them with pairwise comparisons (stating replicate numbers, as said) and with Fig. S2 (left panes). The other statistical results can be given in the form of a table.

Figure 6: Colours in graph do not match colours in legend, presumably because they correspond to species (three shades of blue etc.). But I am afraid this graph also contributes only little: It shows that in terms of the plant traits measured, grasses are more distinct from dicots than the two groups of dicots (legumes and forbs) are from one another. Also, that the two lespedeza species have some overlap and some distinction in these traits. None of this is surprising.

Figure 7: This is what I longed to see: Effect of LC conditioned soil on itself. But as there are no significant soil effects, this graph, too, can go into a table.

Figure 8: So far the most interesting graph in the manuscript. Native legumes have the strongest negative impact on LC biomass production, root length and nodulation, and native forbs the least negative impact – the authors can draw management recommendations from this, and they do; although they might be more specific in exactly how one manages for more native legumes, besides planting them. There are also some interesting contrasts to the native congener L. capitata here: I suggest the authors use these to flesh out their modified filter model, see below.

The discussion is at present too scanty compared to the amount of results presented, and field and greenhouse study had better be discussed together, as the latter might corroborate some speculations from the first. Remaining uncertainties should be discussed openly – they might be phrased as an “outlook” for future studies. The potential impact of LC on non-native species might also be worth a study.

Suggested improvement of the filter model

Consider making this into two graphs with arrows of different sizes indicating “normal” impact by natives; and “abnormal” impact by non-natives. Arrange symbols so that natives are next to the “normal” (slender) arrow and non-natives next to the “abnormal” (fat) arrow. – Use observed differences between LC and L. capitata to prove your point.

Good luck!

Reviewer 2 Report

I think you did a lot of work but you could not present it well. I put some comments in the file. 

When someone reads your work, he/she should be encouraged to continue reading but I can not find it in your work. 

Thanks

Author Response

Reviewer 2 Comments from annotated pdf

**All of this reviewer’s comments were in an annotated pdf of the manuscript that was returned to us. We have extracted these comments and respond to them below in red font following the **.  We thank the reviewer for his/her insightful comments that have helped us improve our manuscript.

Line 229. Figure 2 You can summarize this part in a table. It is difficult to read everything in the text.

** We have moved the vector fitting statistics to Table 1.

Line 332. Figure 6 legend. Long text was used to title almost name of all figures and tables. Why do not you put it in original text before the tables or figures? it is a little confusing for readesr to follow it.

** We have moved the statistics presented in the figure legend into the text (lines 228-232) to make it easier for the reader to follow the figure.

Line 337. Usually discussion and conclusion parts should be your word. But I see a lot of references in following part.

** We do cite a lot of references. We do so to place our interpretation of our results into context of the literature. The wording of the text is very definitely our own and we have not included wording or plagiarism from the references. We have not altered the length of the Discussion to reduce the number of references because the comments by this reviewer about our Discussion contrast with those of Reviewer 1 who suggested that our Discussion was “too scanty”. Nevertheless, we believe that the changes we have made will encourage the reader to continue reading (in response to your comment about this).

Round 2

Reviewer 1 Report

I thank the authors for responding so thoroughly to my comments. I think the manuscript has been significantly improved, and I look forward to seeing it published.

I have very few additional comments and suggestions:

Fig 2 panel (a): Abiotic vectors do not show up sufficiently. Perhaps make them fat and black?

Replication in statistical analysis (page 5 in cover letter): I apologise for not phrasing my concern more clearly in the first place. The points I am trying to make are:

a) I was taught - a while ago -  that there should be about 10 replicates in each tested group (=each bar in your graphs) for robust statistical analysis, that averages should be calculated from minimum 3 replicates, and that replicate number (n) per tested group should be stated in the graphs, or in the text.

In the original design of this study, there are 4 replicates of each species x soil conditioning x competition treatment, which is really very low. It was the right decision to place analyses with such a low number of replication in the supplementary materials only. But it is a good thing they were not omitted entirely, because they prove my second point:

b) When merging/pooling original test groups into a new, larger test group, make sure there is not one original test group that overrides the effects of the others. For example, what appears to be the effect of a given functional group might be largely the effect of one dominant species in that group.

As to number of replicates per tested group:

If all pots could had been included, in Fig 4 there would be 36 replicates per tested group/bar (3 competition treatments x 3 species x 4 replicates); there will in fact be a few less in some groups, but I do not know which exactly.

In Fig 5 there would be 24 replicates per tested group/bar (2 soil conditioning treatments x 3 species x 4 replicates).

In Fig 7 there would also be 24 replicates per tested group/bar (2 soil conditioning treatments x 3 species x 4 replicates), but total sample size here is 144 because the "no competition" treatment is not included - correct?

I suggest to clarify these points.

Good luck!

Author Response

Response to Reviewer 1 (our comments follow ** and are in red font)

I thank the authors for responding so thoroughly to my comments. I think the manuscript has been significantly improved, and I look forward to seeing it published.

** We thank Reviewer 1 for reviewing our manuscript a second time. We are pleased that he/she appreciates the improvements we have made and is now anticipating publication. Below, this reviewer raises two remaining concerns which we respond to. First, he/she suggesting enhancing visibility of the vectors in Fig 2 which we have done. Second, he/she asks for greater clarity in reporting the number of replicates per tested group in Figs 4, 5, and 7; which we have done in the figure legends.

I have very few additional comments and suggestions:

Fig 2 panel (a): Abiotic vectors do not show up sufficiently. Perhaps make them fat and black?

** The color of the vectors was changed to red to make them more visible, while keeping them distinct from the other short black arrows in the figure.

Replication in statistical analysis (page 5 in cover letter): I apologise for not phrasing my concern more clearly in the first place. The points I am trying to make are:

  1. a) I was taught - a while ago -  that there should be about 10 replicates in each tested group (=each bar in your graphs) for robust statistical analysis, that averages should be calculated from minimum 3 replicates, and that replicate number (n) per tested group should be stated in the graphs, or in the text.

In the original design of this study, there are 4 replicates of each species x soil conditioning x competition treatment, which is really very low. It was the right decision to place analyses with such a low number of replication in the supplementary materials only. But it is a good thing they were not omitted entirely, because they prove my second point:

** Thank you for confirming our decisions following your review.

  1. b) When merging/pooling original test groups into a new, larger test group, make sure there is not one original test group that overrides the effects of the others. For example, what appears to be the effect of a given functional group might be largely the effect of one dominant species in that group.

** We appreciate the concern raised by the reviewer here. However, if one dominant species in a group was overriding the effects of others in that group, then we believe that would lead to increased variance associated with that group and a lower likelihood of statistical significance among functional groups. In our analysis of the data we did not see any indication of one species driving the results of a functional groups that would compromise the robustness of our analysis. There were only two instances when there was an interaction between species and treatment (either soil or competition), and in both cases, there was no interaction once the data was analyzed by functional group.

As to number of replicates per tested group:

If all pots could had been included, in Fig 4 there would be 36 replicates per tested group/bar (3 competition treatments x 3 species x 4 replicates); there will in fact be a few less in some groups, but I do not know which exactly.

**We have clarified the specific number of replicates per bar in the Figure descriptions. For Figure 4: Replicates per bar = 36 except unconditioned soil x forbs where n=33, unconditioned soil x grasses where n=33, conditioned soil x forbs where n=33, and conditioned soil x grasses where n=35

In Fig 5 there would be 24 replicates per tested group/bar (2 soil conditioning treatments x 3 species x 4 replicates).

** For Figure 5: Replicates per bar = 24, except no competition x forbs where n=21, no competition x grasses where n=22, L. capitata x forbs where n=23, L. capitata x grasses where n=23, L. cuneata x forbs where n=22, and L. cuneata x grasses =23

In Fig 7 there would also be 24 replicates per tested group/bar (2 soil conditioning treatments x 3 species x 4 replicates), but total sample size here is 144 because the "no competition" treatment is not included - correct?

** For Figure 7: Total n=139. Replicates per bar = 24 except forb x L. capitata where n=23, grass x L. capitata where n=23, forb x L. cuneata where n=22, and grass x L. cuneata where n=23.

I suggest to clarify these points.

Good luck!

** Thank you! We trust that we have now clarified Fig 2 and the sample sizes in Figures 4, 5, & 7 appropriately.

Reviewer 2 Report

Good job!

A lot of work!

For your next paper please work more on the procedure and steps of writing a technical paper. As a reviewer I appreciate all you have done to perform this research. But, as a reader I still have problem to understand your work.

Author Response

Response to Reviewer 2 (our comment follow the ** and are in red font)

Good job!

A lot of work!

For your next paper please work more on the procedure and steps of writing a technical paper. As a reviewer I appreciate all you have done to perform this research. But, as a reader I still have problem to understand your work.

** We thank the reviewer for his/her comments. No new changes have been suggested, so we have not made any additional revisions in response to reviewer 2’s second review.